# TextFlow: A Text Similarity Measure based on Continuous Sequences

## Abstract

Text similarity measures are used in multiple tasks such as plagiarism detection, information ranking and recognition of paraphrases and textual entailment. While recent advances in deep learning highlighted further the relevance of sequential models in natural language generation, existing similarity measures do not fully exploit the sequential nature of language. Examples of such similarity measures include n-grams and skip-grams overlap which rely on distinct slices of the input texts. In this paper we present a novel text similarity measure inspired from a common representation in DNA sequence alignment algorithms. The new measure, called *TextFlow*, represents input text pairs as continuous curves and uses both the actual position of the words and sequence matching to compute the similarity value. Our experiments on eight different datasets show very encouraging results in paraphrase detection, textual entailment recognition and ranking relevance.

## 1 Background

The number of pages required to print the content of the World Wide Web was estimated to 305 billion in a 2015 article[1]. While a big part of this content consists of visual information such as pictures and videos, texts also continue growing at a very high pace. A recent study shows that the average webpage weights 1200KB with plain text accounting for up to 16% of that size[2].

While efficient distribution of textual data and computations are the key to deal with the increasing scale of textual search, similarity measures still play an important role in refining search results to more specific needs such as the recognition of paraphrases and textual entailment, plagiarism detection and fine-grained ranking of information. These tasks also often target dedicated document collections for domain oriented research where text similarity measures can be directly applied.

Finding relevant approaches to compute text similarity motivated a lot of research in the last decades (Sahami and Heilman, 2006; Hatzivassiloglou et al., 1999), and more recently with deep learning methods (Socher et al., 2011; Yih et al., 2011; Severyn and Moschitti, 2015). However, most of the recent advances focused on designing high performance classification methods, trained and tested for specific tasks and did not offer a standalone similarity measure that could be applied (i) regardless of the application domain and (ii) without requiring training corpora.

Yih and Meek (2007) presented an approach to improve text similarity measures for web search queries, with a length ranging from one word to short sequences of words. The proposed method is tailored to this specific task, and the training models are not expected to perform well on different kinds of data such as sentences, questions or paragraphs. Achananuparp et al. (2008) compared several text similarity measures for paraphrase recognition, textual entailment, and the TREC 9 question variants task. In their experiments the best performance was obtained with a linear combination of semantic and lexical similarities, including a word order similarity proposed in (Li et al., 2006) which constructs two vectors of the common words between two sentences and uses their respective positions in the sentences as term weights. The word order similarity measure is then computed by subtracting the two vec-

---
[1] http://goo.gl/p9lt7V
[2] http://goo.gl/c41wpa

tors and taking the absolute value. While such representation takes into account the actual positions of the words, it does not allow detecting subsequence matches and takes into account missing words only by omission.

More generally, existing standalone (or traditional) text similarity measures rely on intersections between token sets, text sizes, and frequency, including measures such as the Cosine similarity, Euclidean distance, Levenshtein, Jaccard and Jaro (Jaro, 1989). The sequential nature of natural language is taken into account mostly through word n-grams and skip-grams which capture distinct slices of the analysed texts but do not preserve the order in which they appear.

In this paper, we use intuitions from a common representation in DNA sequence alignment to design of a new standalone similarity measure called *TextFlow* (*XF*). The proposed measure uses at the same time the full sequence of input texts in a natural sub-sequence matching approach together with individual token matches and mismatches. Our contributions can be detailed further as follows:

- A novel standalone similarity measure which:
    - exploits the full sequence of words in the compared texts.
    - is asymmetric in a way that allows it to provide the best performance on different tasks (e.g., paraphrase detection, textual entailment and ranking).
    - when required, it can be trained with a small set of parameters controlling the impact of sub-sequence matching, position gaps and unmatched words.
    - provides consistent high performance across tasks and datasets compared to traditional similarity measures.

- A neural network architecture to train *TextFlow* parameters for specific tasks.

- An empirical study on both performance consistency and standard evaluation measures, performed with eight datasets from three different tasks.

- A new evaluation measure, called *CORE*, used to better show the consistency of a system at high performance using both its rank

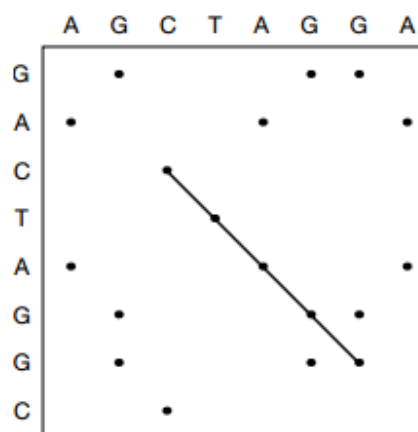

Figure 1: Dot matrix example for 2 DNA sequences (Mount, 2004)

average and rank variance when compared to competing systems over a set of datasets.

## 2 The *TextFlow* Similarity

$XF$ is inspired from a dot matrix representation commonly used in pairwise DNA sequence alignment (cf. figure 2). We use a similar dot matrix representation for text pairs and draw a curve oscillating around the diagonal (cf. figure 2). The area under the curve is considered to be the distance between the two text pairs which is then normalized with the matrix surface. For practical computation, we transform this first intuitive representation using the delta of positions as in figure 3. In this setting, the Y axis is the delta of positions of a word occurring in the two texts being compared. If the word does not occur in the target text, the delta is considered to be a maximum reference value ($l$ in figure 2).

The semantics are: the bigger the area under curve is, the lower the similarity between the compared texts. $XF$ values are real numbers in the [0,1] interval, with 1 indicating a perfect match, and 0 indicating that the compared texts do not have any common token. With this representation, we are able to take into account all matched words and sub-sequences at the same time. The exact value for the $XF$ similarity between two texts $X = \{x_1, x_2, .., x_n\}$ and $Y = \{y_1, y_2, .., y_m\}$ is therefore computed as:

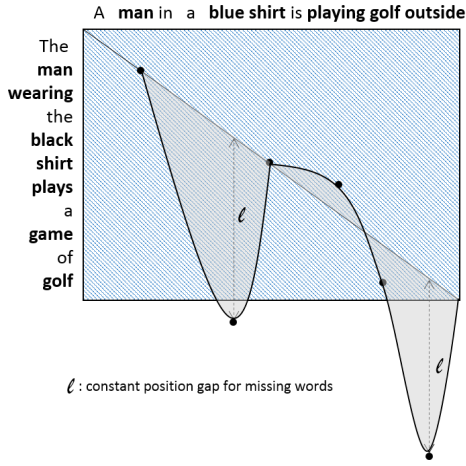

Figure 2: Illustration of *TextFlow* Intuition

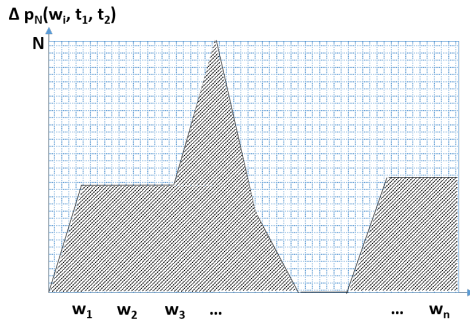

Figure 3: Practical *TextFlow* Computation

$$XF(X,Y) = 1 - \frac{1}{nm}\sum_{i=2}^{n}\frac{1}{S_i}T_{i,i-1}(X,Y)$$
$$- \frac{1}{nm}\sum_{i=2}^{n}\frac{1}{S_i}R_{i,i-1}(X,Y) \quad (1)$$

With $T_{i,i-1}(X,Y)$ corresponding to the triangular area in the $[i-1,i]$ step (cf. figure 3) and $R_{i,i-1}(X,Y)$ corresponding to the rectangular component. They are expressed as:

$$T_{i,i-1}(X,Y) = \frac{|\Delta P(x_i,X,Y) - \Delta P(x_{i-1},X,Y)|}{2} \quad (2)$$

and:

$$R_{i,i-1}(X,Y) = Min(\Delta P(x_i,X,Y), \Delta P(x_{i-1},X,Y)) \quad (3)$$

With:

- $\Delta P(x_i,X,Y)$ the minimum difference between $x_i$ positions in $X$ and $Y$. If $x_i \notin X \cap Y$, $\Delta P(x_i,X,Y)$ is set to the same reference value equal to $m$, (i.e., the cost of a

missing word is set by default to the length of the target text), and:

- $S_i$ is the length of the longest matching sequence between $X$ and $Y$ including the word $x_i$, if $x_i \in X \cap Y$, or 1 otherwise.

$XF$ computation is performed in $O(nm)$ in the worst case where we have to check all tokens in the target text $Y$ for all tokens in the input text $X$. $XF$ is an asymmetric similarity measure. Its asymmetric aspect has interesting semantic applications as we show in the example below (cf. figure 2). The minimum value of $XF$ provided the best differentiation between positive and negative text pairs when looking for semantic equivalence (i.e., paraphrases), the maximum value was among the the top three for the textual entailment example. We conduct this comparison at a larger scale in the evaluation section.

We add 3 parameters to $XF$ in order to represent the importance that should be given to position deltas (Position factor $\alpha$), missing words (sensitivity factor $\beta$), and sub-sequence matching (sequence factor $\gamma$), such as:

$$XF_{\alpha,\beta,\gamma}(X,Y) = 1 - \frac{1}{\beta nm}\sum_{i=2}^{n}\frac{\alpha}{S_i^{\gamma}}T_{i,i-1}^{\beta}(X,Y)$$
$$- \frac{1}{\beta nm}\sum_{i=2}^{n}\frac{\alpha}{S_i^{\gamma}}R_{i,i-1}^{\beta}(X,Y) \quad (4)$$

With:

$$T_{i,i-1}^{\beta}(X,Y) = \frac{|\Delta_{\beta}P(x_i,X,Y) - \Delta_{\beta}P(x_{i-1},X,Y)|}{2} \quad (5)$$

$$R_{i,i-1}^{\beta}(X,Y) = Min(\Delta_{\beta}P(x_i,X,Y), \Delta_{\beta}P(x_{i-1},X,Y)) \quad (6)$$

and:

- $\Delta_{\beta}P(x_i,X,Y) = \beta m$, if $x_i \notin X \cap Y$

- $\alpha < \beta$: forces missing words to always cost more than matched words.

- $S_i^{\gamma} = \begin{cases} 1 \ if \ S_i = 1 \ or \ x_i \notin X \cap Y \\ \gamma \ S_i \ for \ S_i > 1 \end{cases}$

The $\gamma$ factor increases or decreases the impact of sub-sequence matching, $\alpha$ applies to individual token matches whether inside or outside a sequence, and $\beta$ increases or decreases the impact of

| Positive Entailment | $E_1$ | Under a blue sky with white clouds, a child reaches up to touch the propeller of a plane standing parked on a field of grass. |
| | $E_2$ | A child is reaching to touch the propeller of a plane. |
| Negative Entailment | $E_3$ | Two men on bicycles competing in a race. |
| | $E_4$ | Men are riding bicycles on the street. |
| Positive Paraphrase | $P_1$ | The most serious breach of royal security in recent years occurred in 1982 when 30-year-old Michael Fagan broke into the queen's bedroom at Buckingham Palace. |
| | $P_2$ | It was the most serious breach of royal security since 1982 when an intruder, Michael Fagan, found his way into the Queen's bedroom at Buckingham Palace. |
| Negative Paraphrase | $P_3$ | "Americans don't cut and run, we have to see this misadventure through," she said. |
| | $P_4$ | She also pledged to bring peace to Iraq: "Americans don't cut and run, we have to see this misadventure through." |

| Task | Entailment Recognition | | | Paraphrase Detection | | |
|---|---|---|---|---|---|---|
| Sentence Pair | $(E_1, E_2)$ | $(E_3, E_4)$ | $(E_1, E_2)$ - $(E_3, E_4)$ | $(P_1, P_2)$ | $(P_3, P_4)$ | $(P_1, P_2)$ - $(P_3, P_4)$ |
| Example class (Pos/Neg) | (Pos) | (Neg) | (Gap) | (Pos) | (Neg) | (Gap) |
| Jaro-Winkler | 0.629 | 0.712* | -0.083** | **0.884** | 0.738 | 0.146 |
| Levenshtein | 0.351 | 0.259 | 0.092 | 0.708 | 0.577 | 0.130 |
| Jaccard | 0.250* | **0.143** | 0.107 | 0.571* | 0.583 | -0.012 |
| Cosine | 0.462 | 0.250 | 0.212 | 0.730 | 0.746** | -0.016 |
| Word Overlap | **0.800** | 0.250 | **0.550** | 0.800 | 0.875* | -0.075 |
| MIN(XF (x,y), XF(y,x)) | 0.260** | 0.563** | -0.303* | 0.693** | **0.497** | **0.196** |
| MAX(XF(x,y), XF(y,x)) | 0.707 | 0.563** | 0.144 | 0.832 | 0.739 | 0.093 |

Figure 4: Example sentences and similarity values. The best value per column is highlighted. The second best is underlined. Worst and second worst values are followed by one and two stars. Entailment examples are taken from SNLI (Bowman et al., 2015). Paraphrase examples are taken from MSRP [4].

missing tokens as well as the normalization quantity $\beta nm$ in equation 4 to keep the similarity values in the [0,1] range.

## 2.1 Parameter Training

By default *XF* has canonical parameters set to 1. However, when needed, $\alpha$, $\beta$, and $\gamma$ can be trained on learning data for a specific task. We designed a neural network to perform this task, with a hidden layer dedicated to compute the exact XF value. To do so we compute, for each input text pair, the coefficients vector that would lead exactly to the XF value when multiplied by the vector $< \frac{\alpha}{\beta}, \frac{\alpha}{\beta\gamma}, \beta >$. Figure 5) presents the training neural network considering several types of sequences (or translations) of the input text pairs (e.g., lemmas, words, synsets).

We use identity as activation function in the dedicated XF layer in order to have a correct comparison with the other similarity measures, including canonical XF where the similarity value is provided in the input layer (cf. figure 6).

## 3 Evaluation

**Datasets**. This evaluation was performed on 8 datasets from 3 different classification tasks: Textual Entailment Recognition, Paraphrase Detection, and ranking relevancy. The datasets are as follows:

- **RTE 1, 2, and 3**: the first three datasets from the Recognizing Textual Entailment (RTE) challenge (Dagan et al., 2006). Each dataset consists of sentence pairs which are annotated with 2 labels: entailment, and non-entailment. They contain respectively (200, 800), (800, 800), and (800, 800) (train, test) pairs.

- **Guardian**: an RTE dataset collected from 78,696 Guardian articles[5] published from January 2004 onwards and consisting of 32K pairs which we split randomly in 90%/10% training/test sets. Positive examples were collected from the titles and first sentences. Negative examples were collected from the same source by selecting consecutive sentences and random sentences.

- **SNLI**: a recent RTE dataset consisting of 560K training sentence pairs annotated with

---

[5] https://github.com/daoudclarke/rte-experiment

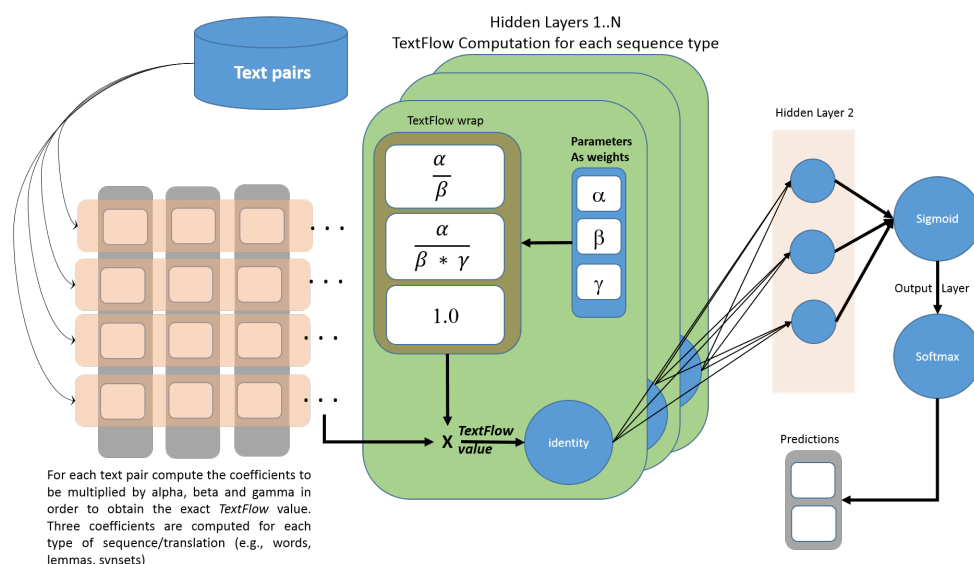

Figure 5: NN architecture **A1** for XF Parameter Training

3 labels: entailment, neutral and contradiction (Bowman et al., 2015). We discarded the contradiction pairs as they do not necessarily represent dissimilar sentences and are therefore a random noise w.r.t. our similarity measure evaluation.

- **MSRP**: the Microsoft Research Paraphrase corpus, consisting of 5,800 sentence pairs annotated with a binary label indicating whether the two sentences are paraphrases or not.

- **Semeval-16-3B**: a dataset of question-question similarity collected from Stack-Overflow (Nakov et al., 2016). The dataset was contains 3,169 training pairs and 700 test pairs. Three labels are considered: "Perfect Match", "Relevant" or "Irrelevant". We combined the first two into the same positive category for our evaluation.

- **Semeval-14-1**: a corpus of Sentences Involving Compositional Knowledge (Marelli et al., 2014) consisting of 10,000 english sentence pairs annotated with both similarity scores and relevance labels.

**Features.** After a preprocessing step where we removed stopwords, we computed the similarity values using 7 different types of sequences constructed, respectively, with the following value from each token:

- Word (plain text value)

- Lemma

- Part-Of-Speech (POS) tag

- WordNet Synset[6] OR Lemma

- WordNet Synset OR Lemma for Nouns

- WordNet Synset OR Lemma for Verbs

- WordNet Synset OR Lemma for Nouns and Verbs.

In the 4 last types of sequences the lemma is used when there is no corresponding WordNet synset. In a first experiment we compare different aggregation functions on top of XF (i.e., minimum, maximum and average) in table 3. We used the LibLinear[7] SVM classifier for this task.

In the second part of the evaluation, we use neural networks to compare the efficiency of $XF_c$, $XF_t$ and other similarity measures with in the same setting. We use the neural net described in figure 5 for the trained version $XF_t$ and the equivalent architecture presented in figure 6 for all other similarity measures. For canonical $XF_c$ we use by default the best aggregation for the task as observed in table 3.

---

[6] https://wordnet.princeton.edu/
[7] https://www.csie.ntu.edu.tw/~cjlin/liblinear/

| Task | Entailment Recognition | | | | | Paraphrase Detection | Ranking Relevance | |
|---|---|---|---|---|---|---|---|---|
| Datasets | RTE 1 | RTE 2 | RTE 3 | Guardian | SNLI | MSRP | Semeval16-t3B | Semeval12-t17 |
| XF MIN | **55.3** | 53.8 | 60.0 | 77.3 | 58.0 | **72.1** | 77.4 | 77.8 |
| XF AVG | 51.4 1 | 57.2 | 62.5 | 84.9 | 62.0 | 72.0 | **77.6** | **79.5** |
| XF MAX | 53.9 | **61.3** | **64.7** | **86.7** | **64.3** | 71.4 | 76.7 | 77.7 |

Table 1: Accuracy evaluation with different aggregations of XF using an SVM classifier.

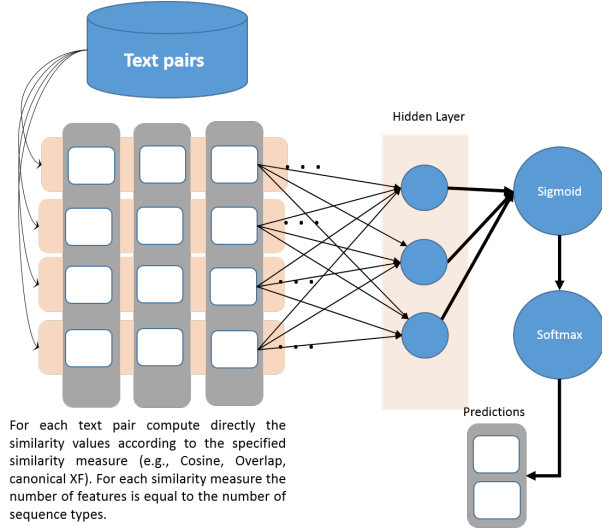

For each text pair compute directly the similarity values according to the specified similarity measure (e.g., Cosine, Overlap, canonical XF). For each similarity measure the number of features is equal to the number of sequence types.

Figure 6: NN Architecture **A2** for the equivalent evaluation of other similarity measures.

**Similarity Measures**. We considered nine traditional similarity measures included in the Simmetrics distribution[8]: Cosine, Euclidean distance, Word Overlap, Dice coefficient (Dice, 1945), Jaccard(Jain and Dubes, 1988), Damerau, Jaro-Winkler (JW) (Porter et al., 1997), Levenshtein (LEV) (Sankoff and Kruskal, 1983), and Longest Common Subsequence (LCS) (Friedman and Sideli, 1992).

**Implementation.** XF was implemented in Java as an extension of the Simmetrics package, made available at this address[9]. The neural neworks were implemented in Python with TensorFlow[10]. We also share the training sets used for both parameter training and evaluation. The evaluation was performed on a 4-cores laptop with 32GB of RAM. The initial parameters for $XF_t$ were chosen with a random function.

**Evaluation Measures.** We use the standard accuracy values and F1, precision and recall for the

---

[8] https://github.com/Simmetrics/simmetrics

[9] Code link omitted for blind review.

[10] https://www.tensorflow.org/

positive class (i.e., entailment, paraphrase, and ranking relevance). We also study the relative rank in performance of each similarity measure across all datasets using the average rank, the rank variance and a new evaluation measure called cOnsistent peRformancE (CORE), computed as follows for a system $m$, a set of datasets $D$, a set of systems $S$, and an evaluation measure $E \in \{F1, Precision, Recall, Accuracy\}$:

$$CORE_{D,S,E}(m) = \frac{\underset{p \in S}{MIN}\left(\underset{d \in D}{AVG}(R_S(E_d(p)) + V_{d \in D}(R_S(E_d(p))))\right)}{\underset{d \in D}{AVG}\left(R_S(E_d(m))\right) + V_{d \in D}\left(R_S(E_d(m))\right)} \quad (7)$$

With $R_S(E_d(m))$ the rank of $m$ according to the evaluation measure $E$ on dataset $d$ w.r.t. competing systems $S$. $V_{d \in D}(R_S(E_d(m)))$ is the rank variance of $m$ over datasets. The intuition can be read on the results tables 2, 3, and 4. Basically, $CORE$ tells us how consistent a system/method is in having high performance, relatively to the set of competing systems $S$. The maximum value of $CORE$ is 1 for the best performing system according to its rank. It also allows quantifying how consistently a system achieves high performance for the remaining systems.

Tensorflow outperformed the results obtained with a combination of word order similarity and semantic similarities tested in (Achananuparp et al., 2008), with gaps of +1.0 in F1 and +6.1 accuracy on MSRP and +4.2 F1 and +2.7% accuracy on RTE 3.

# 4 Discussion

**Canonical *Text Flow*.** $TF_c$ had the best average and micro-average accuracy on the 8 classification datasets, with a gap of +0.4 to +6.3 when compared to the state-of-the-art measures. It also reached the best precision average with a gap of +1.8 to +6.3. On the F1 score level $XF_c$ achieved the second best performance with a gap of -1.7, mainly caused by its underperformance in recall, where it had the third best performance with a gap of -6.3. The recall table is omitted due to a lack

| | Cosine | Euc | Overlap | Dice | Jaccard | Damerau | JW | LEV | LCS | $XF_C$ | $XF_T$ |
|---|---|---|---|---|---|---|---|---|---|---|---|
| RTE 1 | .561 | .564 | .550 | .504 | .511 | .557 | .532 | .561 | _.568_ | .550 | **.575** |
| RTE 2 | .575 | .555 | _.598_ | .566 | .572 | .548 | .541 | .551 | .548 | .597 | **.612** |
| RTE 3 | **.652** | .562 | .636 | .637 | .630 | .567 | .538 | .567 | .562 | .627 | _.647_ |
| Guardian | .748 | .750 | .820 | .778 | .780 | .847 | .726 | .847 | .848 | _.867_ | **.876** |
| SNLI | .621 | .599 | **.665** | .612 | .608 | .631 | .556 | .630 | .619 | .641 | _.656_ |
| MSRP | .719 | .689 | .720 | _.729_ | .731 | .687 | .699 | .685 | .717 | .724 | **.732** |
| Semeval-16-3B | .756 | .734 | .769 | _.781_ | .780 | .759 | .751 | .759 | .737 | .777 | **.782** |
| Semeval-14-1 | _.790_ | .756 | .779 | .783 | .786 | .749 | .719 | .749 | .757 | .783 | **.798** |
| AVG | .678 | .651 | .692 | .674 | .675 | .668 | .633 | .669 | .670 | _.696_ | **.710** |
| Micro Avg | .699 | .675 | .725 | .700 | .700 | .701 | .646 | .701 | .701 | _.726_ | **.739** |
| RANK Avg | 5.1 | 8.2 | 4.5 | 5.6 | 5.5 | 6.9 | 10.1 | 6.7 | 6.7 | _4.1_ | **1.2** |
| RANK Var. | 9.0 | 5.9 | 4.3 | 10.0 | 8.6 | 5.3 | 1.6 | 6.2 | 8.2 | _2.7_ | **0.2** |
| CORE | 0.104 | 0.103 | 0.167 | 0.094 | 0.104 | 0.121 | 0.125 | 0.113 | 0.098 | _0.215_ | **1.000** |

Table 2: Accuracy values using. The best result is highlighted, the second best is underlined.

| | Cosine | Euc | Overlap | Dice | Jaccard | Damerau | JW | LEV | LCS | $XF_C$ | $XF_T$ |
|---|---|---|---|---|---|---|---|---|---|---|---|
| RTE 1 | _.612_ | .564 | **.636** | .512 | .523 | .578 | .513 | .583 | .494 | .565 | .599 |
| RTE 2 | .579 | .590 | **.662** | .565 | .558 | .549 | .516 | .551 | .555 | .616 | _.646_ |
| RTE 3 | **.705** | .598 | .682 | _.695_ | .682 | .608 | .556 | .607 | .603 | .665 | .690 |
| Guardian | .742 | .749 | .816 | .774 | .776 | .849 | .713 | .849 | .850 | _.862_ | **.873** |
| SNLI | _.582_ | .576 | **.641** | .562 | .564 | .627 | .479 | .627 | .611 | .594 | _.585_ |
| MSRP | .808 | .797 | .812 | **.814** | _.813_ | .784 | .802 | .783 | .804 | .804 | .810 |
| Semeval-16-3B | .632 | .462 | .625 | .648 | .644 | .544 | .545 | .547 | .508 | .633 | **.662** |
| Semeval-14-1 | .764 | .707 | .748 | .753 | .746 | .706 | .680 | .706 | .714 | .744 | .673 |
| AVG | .678 | .630 | **.702** | .665 | .663 | .655 | .600 | .656 | .642 | .685 | _.692_ |
| Micro Avg | .684 | .656 | **.716** | .679 | .677 | .691 | .608 | .692 | .688 | _.702_ | .687 |
| RANK Avg | _4.5_ | 8.12 | 3.12 | 5.12 | 5.5 | 6.89 | 9.88 | 6.62 | 7.12 | 4.62 | **3.88** |
| RANK Var. | 9.7 | _4.7_ | 4.4 | 14.7 | 6.6 | 8.7 | 1.8 | 9.1 | 8.1 | **2.3** | 11.0 |
| CORE | 0.485 | 0.538 | 0.915 | 0.348 | _0.571_ | 0.443 | 0.588 | 0.438 | 0.452 | **1.000** | 0.464 |

Table 3: F1 scores. The best result is highlighted, the second best is underlined.

| | Cosine | Euc | Overlap | Dice | Jaccard | Damerau | JW | LEV | LCS | $XF_C$ | $XF_T$ |
|---|---|---|---|---|---|---|---|---|---|---|---|
| RTE 1 | .548 | .564 | .534 | .503 | .510 | .552 | .535 | .555 | **.596** | .546 | _.566_ |
| RTE 2 | .574 | .547 | .571 | .567 | .578 | .547 | .546 | .551 | .546 | _.588_ | **.594** |
| RTE 3 | .624 | .565 | _.618_ | .611 | .610 | .568 | .547 | .568 | .564 | .616 | **.627** |
| Guardian | .759 | .753 | .836 | .789 | .789 | .839 | .749 | .840 | .839 | _.891_ | **.894** |
| SNLI | .644 | .608 | _.690_ | .642 | .632 | .631 | .577 | .630 | .621 | .679 | **.735** |
| MSRP | .740 | .705 | .732 | .749 | .755 | .723 | .713 | .722 | .743 | _.760_ | **.765** |
| Semeval-16-3B | .634 | **.708** | .678 | .698 | .698 | .732 | .698 | .729 | .674 | _.700_ | .686 |
| Semeval-14-1 | .745 | .738 | .738 | .743 | **.769** | .716 | .672 | .716 | .727 | _.762_ | .740 |
| AVG | .659 | .649 | .675 | .663 | .668 | .664 | .630 | .664 | .664 | _.693_ | **.701** |
| Micro Avg | .693 | .674 | .721 | .699 | .704 | .694 | .645 | .693 | .693 | _.737_ | **.752** |
| RANK Avg. | 5.6 | 7.5 | 5.9 | 5.9 | 5.1 | 6.1 | 9.6 | 6.1 | 7.1 | _3.2_ | **2.5** |
| RANK Var. | 9.4 | 10.0 | 6.4 | 5.3 | 7.8 | 7.0 | _4.6_ | 7.6 | 11.6 | **3.1** | 6.9 |
| CORE | 0.420 | 0.361 | 0.515 | 0.567 | 0.488 | 0.482 | 0.446 | 0.462 | 0.338 | **1.000** | _0.676_ |

Table 4: Precision values. The best result is highlighted, the second best is underlined.

of space. On a rank level, $XF_c$ had the best consistent rank for accuracy F1 and precision, and the second best for recall.

**Trained *Text Flow*.** When compared to soa measures and to canonical XF, the trained version, $XF_t$, obtained the best accuracy with a gap ranging from +1.4 to +7.8. $XF_t$ also obtained the second best F1 average with a -1.0 gap, but with clear inconsistencies according to the dataset. $XF_t$ obtained the best precision with a gap ranging from +0.8 to +7.1. $XF_t$ did not perform well on recall with 64.5% micro-average compared to WordOverlap with 72%. Both its recall and F1 performance can be explained by the fact that the measure was trained to optimize accuracy, and not the F1 score for the positive class; which also suggests that the approach could be adapted to F1 optimization if needed. Canonical XF was more consistent than trained XF on all dimensions except accuracy, for which $XF_t$ was optimized. We argue that this consistency was made possible through the asymmetry of XF which allowed it to adapt to different kinds of similarities (i.e., equivalence/paraphrase, inference/entailment, and mutual distance/ranking). These results also show that the actual positions difference is a relevant factor for text similarity. We explain it mainly by the natural flow of language where the important entities and relations are often expressed first, in contrast with a purely logical-driven approach which has to consider, for instance, that active forms and passive forms are equivalent in meaning. The difference in positions is also not read literally, both because of the higher impact associated to missed words and to the $\alpha$ parameter which allows leveraging their impact in the trained version.

In additional experiments, we compared $TF_c$ and $TF_t$ with the other similarity measures when applied to bi-grams and tri-grams instead of individual tokens. The results were significantly lower on all datasets (between 3 and 10 points loss in accuracy) for both the soa measures and *TextFlow* variants. This result could be explained by the fact that n-grams are too rigid when a sub-sequence varies even slightly, e.g., the insertion of a new word inside a 3-words sequence leads to a tri-gram mismatch and reduces bi-gram overlap from 100% to 50% for the considered sub-sequence. This issue is not encountered with TextFlow as it relies on the token level, and such an insertion will

not cancel or reduce significantly the gains from the correct ordering of the words. We also conducted ranking correlation experiments on three test datasets provided at the semantic text similarity task at Semeval 2012, with gold score values for their text pairs. The datasets have 750 sentence pairs each, and are extracted from the Microsoft Research video descriptions corpus, MSRP and the SMTeuroparl[11]. When compared to the traditional similarity measures, *TextFlow* had the best correlation on the first two datasets with, for instance, 0.54 and 0.46 pearson correlation on the lemmas sequences and the second best on the MSRP extract where the Cosine similarity had the best performance with 0.71 vs 0.68, noting that the Cosine similarity uses word frequencies when the evaluated version of TextFlow did not use word-level weights.

Including word weights is one of the promising perspectives in line with this work as it could be done simply by making the deltas vary according to the weight/importance of the (un)matched word. Also, in such setting, the impact of a sequence of $N$ words will naturally increase or decrease according to the word weights (cf. figure 3). We conducted a preliminary test using the inverse document frequency of the words as extracted from Wikipedia with Gensim[12], which led to an improvement of up to 2% for most datasets, with performance decreasing slightly on two of them. Other kinds of weights could also be included just as easily, such as contextual word relatedness using embeddings or other semantic relatedness factors such as WordNet distances (Pedersen et al., 2004).

## 5 Conclusion

We presented a novel standalone similarity measure that takes into account continuous word sequences. Evaluation on eight datasets show promising results for textual entailment recognition, paraphrase detection and ranking. Among the potential extensions of this work are the inclusion of different kinds of weights such as TF-IDF, embedding relatedness and semantic relatedness. We also intend to test other variants around the same concept, including considering the matched words and sequences to have a negative weight to balance further the weight of missing words.

---

[11]goo.gl/NVnybD
[12]https://radimrehurek.com/gensim/

## Acknowledgements

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
