# Peer review of "TextFlow: A Text Similarity Measure based on Continuous Sequences"

_ACL 2017 — decision unknown_

[Official Review · Reviewer 1 · rating 4 · confidence 3]
soundness 4 · originality 4 · clarity 5 · impact 3 · substance 4 · appropriateness 5 · meaningful comparison 4 · presentation format Oral Presentation

- Strengths:

originality of the CORE evaluation measure, good accuracy of proposed
similarity measure and large number and diversity of datasets for evaluation.

- Weaknesses: 

 # some typos
   - line 116-117, 'to design of a new' -> 'to design a new'
   - line 176-177, figure 2 -> figure 1
   - line 265, 'among the the top' -> 'among the top'
   - line 320, 'figure 4' should be introduced within the article body.
   - line 434, 'the dataset was contains' -> 'the dataset contains'
   - line 486-487, table 3 -> table 1
   - a 'Tensorflow' should be replaced by 'TextFlow'

 # imprecisions
   - features computation accuracy of lemma, pos or wordnet synset should be
detailed in the paper and it should be discussed if it impacts the general
similarity accuracy evaluation or not
  - the neural networks are said to be implemented in Python but the code is
not
said to be available - to be able to repeat the experiment
  - the training and evaluation sets are said to be shared, but it is not said
how (on demand?, under license?) - to be able to repeat the experiment

- General Discussion: